# Charged Particles Transverse Momentum and Pseudorapidity Distribution in Hadronic Collisions at LHC Energies

**DOI:** 10.3390/e25030452

**Published:** 2023-03-04

**Authors:** Muhammad Ajaz, Abd Al Karim Haj Ismail, Mateen Ullah Mian, Rashid Khan, Ramoona Shehzadi, Muhammad Adil Khan, Atef AbdelKader, Muhammad Waqas, Elmuez A. Dawi, Uzma Tabassam

**Affiliations:** 1Department of Physics, Abdul Wali Khan University Mardan, Mardan 23200, Pakistan; 2College of Humanities and Sciences, Ajman University, Ajman P.O. Box 346, United Arab Emirates; 3Nonlinear Dynamics Research Center (NDRC), Ajman University, Ajman P.O. Box 346, United Arab Emirates; 4Department of Physics, Islamia College Peshawar, Peshawar 25120, Pakistan; 5Department of Physics, University of the Punjab, Lahore 54590, Pakistan; 6School of Mathematics, Physics and Optoelectronic Engineering, Hubei University of Automotive Technology, Shiyan 442002, China; 7Department of Physics, COMSATS University Islamabad, Islamabad 44000, Pakistan

**Keywords:** Monte-Carlo models, transverse momentum, pseudorapidity, statistical models, effective temperature

## Abstract

We present an analysis of the pseudorapidity η and transverse momentum $p_{T}$ distributions of charged hadrons in pp collisions for the kinematic range of $0<p_{T}<4$ GeV/c and |η|<2.4 at 0.9, 2.36, and 7 TeV. Charged particles are produced in pp collision using several Monte Carlo event generators (Pythia Simple, Vincia, Dire showers, Sibyll2.3d, QGSJETII-04, EPOS-LHC) and compared with CMS data at LHC. It is observed that the Simple parton showers can explain the CMS data very well for $p_{T}>1$ GeV/c at 0.9 and 2.36 TeV within the experimental errors, while Dire overshoots and Vicia undershoots the data by 50% each. At 7 TeV, the Dire module presents a good prediction, whereas the Simple and Vincia modules underestimate the data within 30% and 50%. Comparing the Simple module of the Pythia model and the predictions of the CRMC models with the experimental data shows that at 0.9 TeV, EPOS-LHC has better results than the others. At 2.36 GeV, the cosmic rays Monte Carlo (CRMC) models have better prediction than the Simple module of Pythia at low $p_{T}$, while QGSJETII-04 predicts well at high $p_{T}$. QGSJETII-04 and EPOS-LHC have closer results than the Pythia-Simple and Sibyll2.3d at 7 TeV. In the case of the pseudorapidity distributions, only the Pythia-Simple reproduced the experimental measurements at all energies. The Dire module overestimates, while Vincia underestimates the data in decreasing order of discrepancy (20%, 12%, 5%) with energy. All CRMC models underestimate the data over the entire η range at all energies by 20%. The angular ordering of partons and the parton fragmentation could be possible reasons for this deviation. Furthermore, we used the two-component standard distribution to fit the $p_{T}$ spectra to the experimental data and extracted the effective temperature ($T_{eff}$) and the multiplicity parameter ($N_{0}$). It is observed that $T_{eff}$ increases with the increase in the center of mass energy. The fit yielded 0.20368±0.01, 0.22348±0.011, and 0.24128±0.012 GeV for 0.9, 2.36, and 7 TeV, respectively. This shows that the system at higher energies freezes out earlier than lower ones because they quickly attain the equilibrium state.

## 1. Introduction

It is essential to measure particle yields and kinematic distributions to explore new energy regimes of hadronic collisions. Such studies provide insights into the soft and hard scattering contributions and a deep understanding of the hadron production, which in its turn provides a better platform for new investigations [1,2,3]. For the pp collisions at LHC, an intensive knowledge of inclusive particle production will make it easy to distinguish the background from the signal in soft hadronic interactions. Furthermore, the pp study provides a baseline for the heavy systems by giving information on nuclear medium effects [4,5]. In pp collisions, soft interactions produce most particles and can be modeled phonologically. The most widely used models and event generators can be tuned with the help of comparison to experimental data. These soft collisions include the inelastic single diffractive (ND), non-single diffractive (NSD), double diffractive (DD), and elastic scatterings [6].

The current study aims to contrast the prediction of event generators using the transverse momentum ($p_{T}$) and pseudorapidity (η) distributions of charged particles at several LHC energies. Such parameters are key observables to distinguish general properties of the collision that are more significant for higher collision energies [7,8]. Similar studies showing models’ prediction compared to experimental data at different energies are given in the references [9,10,11,12,13,14,15,16,17,18,19,20]. The data at 0.9 and 2.36 TeV are taken from [4] and 7 TeV from [5].

The rest of the paper is organized as follows: Section 2 explains the methods adopted for this work. Section 3 contains a detailed discussion of the obtained results, and Section 4 contains concluding remarks.

## 2. The Method and Formalism

For the production of charged hadrons, mini bias Monte Carlo event generators, including PYTHIA8 and parton showers, EPOS-LHC, QGSJETII-04, and Sibyll2.3d, have been used, and the obtained results are contrasted against the CMS data at LHC. For the simulation purpose, the definitions of the kinematic observable have been used in the models. In the momentum space, the transverse momentum is defined as follows: $p_{T}^{2}=p_{x}^{2}+p_{y}^{2}$; where $p_{x}^{2}$ is the momentum of the particles along x-axis and $p_{y}^{2}$ is the momentum of the particles along y-axis. For the pseudorapidity measure, the definition of η=−lntan(θ/2) is used. A total of five hundred thousand events are generated in each case.

PTHIA8 is a general-purpose Monte Carlo event generator for heavy ion hadronic collisions. PYTHIA8 rests on the physics concepts of soft and hard processes, multi-parton interactions, parton showers, and string fragmentation. It also links the parton showers, such as Dire showers, Vincia showers, and Simple showers (TimeShowers and SpaceShowers). All three shower models perform ordinary QCD and QED radiation, whereas parton showers provide the baseline for linking external showers [21,22]. In order to tune PYTHIA8 to different parton showers, the selection of mode from 1 to 3 is chosen as PartonShowers:modelmode=1,2,3. For mode = 1, the Simple parton shower is activated; for mode = 2, the Vincia shower is activated; for mode = 3, the Dire shower is activated [23,24,25]. A description of the models’ prediction and their comparison to the experimental data in this study is given below.

Cosmic-ray Monte Carlo (CRMC) models (EPOS-LHC, Sibyll2.3d, and QGSJETII-04) are minimum-biased event generators used for the hadron production in hadronic and nuclear interactions at high energies. Moreover, cosmic ray models are very well suited and flexible to describe the hadronic bulk particle production; therefore, they can be used at LHC energies for efficiency corrections. The production of events is very trivial With CRMC models [7,26]. These models are based on Gribov’s Reggeon Field Theory (RFT) [27]. It works on the production of particles, and studies the cross-section via the exchange of Pomerons and Reggeons. At higher energies, the contribution of Pomeron dominates through the exchange of multi-gluons and those from secondary Reggeons. Notably, the exchange of soft Pomerons, which are of colorless nature, is mainly responsible for the production of particles in non-central hadronic collisions and diffractive dissociations. The latter is responsible for the total inelastic cross-section up to asymptotic energies.

All the CRMC models are based on soft and (semi)-hard processes of particle production by considering the multi-Pomeron interactions. As a result of high density of the partons at small values of *x*, the use of the remnants hadrons is different in different generators. The Sibyll2.3d model makes use of an energy-dependent transverse momentum cut-off for the production of mini-jet and relies on the less number of gluons in a hadron for given transverse area, While in EPOS-LHC, the elastic and inelastic fusion of parton-ladders is included to address the nonlinear phenomenological effects [28].

Besides comparing the experimental measurements with the predictions of models, we used the standard distribution to reproduce the experimental data and extract the effective temperature $T_{eff}$ of the produced medium. The standard distribution is chosen considering the temperature has the closest concept as in the ideal gas model, which includes the chemical potential (μ) and the index of standard distribution (*S*). This means that the $p_{T}$-dependent probability density function has the form [29]:(1)$$\begin{array}{cc} f_{p_{T}}\left(p_{T},T\right) & =\frac{1}{N}\frac{dN}{dp_{T}}=Cp_{T}m_{T}\int_{y_{minimum}}^{y_{maximum}}\cosh y\\ \; & \times {\left[\exp\left(\frac{m_{T}\cosh y-\mu }{T}\right)\pm S\right]}^{-1}dy, \end{array}$$ In Equation (Equation 1), the transverse mass ($m_{T}$) is given by
$$\begin{array}{c} m_{T}=\sqrt{p_{T}^{2}+m_{0}^{2}}, \end{array}$$
where $m_{0}$ is used to represent the rest-mass, and *N* shows the particles’ multiplicity. The minimum and maximum rapidity are represented by $y_{minimum}$ and $y_{maximum}$, respectively. *S* has a value of 1, while the −ve and +ve values are for bosons and fermions, respectively. Finally, *C* is used as a constant of normalization, which gives the integral of Equation (Equation 1) to be normalized to unity.

The parameter μ in Equation (Equation 1) is particle-dependent, where *i* is a particle type. The following equation can show the chemical potential ($\mu_{i}$) of these particles by [30,31,32]:(2)$$\begin{array}{c} \mu_{i}=-\frac{1}{2}T_{ch}\ln\left(z_{i}\right), \end{array}$$ In Equation (Equation 2), $z_{i}$ represents the negative to positive particles’ ratio, while $T_{ch}$ in the last equation is the temperature at the chemical freeze-out stage in the statistical thermal model [33,34,35,36,37] and is given by the following expression:(3)$$\begin{array}{c} T_{ch}=\frac{T_{limit}}{1+\exp\left[2.60-\ln\left(\sqrt{s_{NN}}\right)/0.450\right]}, \end{array}$$
where $T_{limit}=0.1580$ GeV is the saturation or limiting temperature [38], and $\sqrt{s_{NN}}$ has the usual meaning of the cms energy in GeV units.

One may need to use standard distributions with two-component to fit the $p_{T}$ spectra. A special case is when there is a contribution from the decay of resonances; then, a two-component distribution for a good fit is required. Similarly, if hard scattering has a considerable contribution in the $p_{T}$ region under consideration, a two-component distribution will be a good choice. In the former case, where the decay of resonances contributes is covering a very low-$p_{T}$ region of the spectra (<0.20∼0.30 GeV/*c*), the total probability density function of the $p_{T}$ of the two components can be written in the form of the superposition principle as follows:(4)$$\begin{array}{c} f_{p_{T}}\left(p_{T}\right)=zf_{1}\left(p_{T},T_{1}\right)+\left(1-z\right)f_{2}\left(p_{T},T_{2}\right) \end{array}$$ In Equation (Equation 4), *z* is the contribution fraction coming from the first component ($f_{1}\left(p_{T},T_{1}\right)$ to [$f_{2}\left(p_{T},T_{2}\right)$]). The integral of Equation (Equation 4) is normalized to one. Hence,
(5)$$\begin{array}{c} T=zT_{1}+\left(1-z\right)T_{2} \end{array}$$ Equation (Equation 5) is the weighted average of the two components. The temperature (*T*) given in Equation (Equation 5) is the $T_{eff}$, assuming both components stay in equilibrium.

## 3. Results and Discussions

We investigate the transverse momentum ($p_{T}$) and pseudorapidity (η) distributions of primary charged hadrons in the kinematic range of $0<p_{T}<4$ GeV/c and |η|<2.4 at 0.9, 2.36, and 7 TeV. The minimum biased Monte Carlo (MC) event generators, i.e., PYTHIA8.307, EPOS-LHC, QGSJETII-04, and Sibyll2.3d, are used to produce primary charged hadrons, and then the predictions are compared to the CMS data at 0.9 TeV, 2.36 TeV [4], and 7 TeV [5] at the LHC. Since there are three different modules in the PYTHIA8 for parton showers, Simple (Timelike, Spacelike), Vincia, and Dire showers, the predictions of these three modules are first compared to the experimental data [4,5]. The best of the three modules is then used with other MC generators for comparison with the experimental data. A description of the models’ prediction compared to the data is given in the following section.

### 3.1. Comparison of Models’ Prediction with Data

A detailed description of the model’s predictions compared to the CMS data at LHC is outlined here. Figure 1 shows the distributions of the $p_{T}$ of all charged hadrons at the three CMS energies within the kinematic range mentioned above. It is worth mentioning that filled black circles show the experimental data, while lines of different colors represent the model results. The left panels of the figure compare the three modules of Pythia with the experimental data, where red, blue, and green colors are used to display the predictions of the Pythia-Simple, Pythia-Vincia, and Pythia-Dire, respectively. The right panels compare CRMC models and Pythia (Simple module) with the experimental data, where red, blue, green, and yellow show the results of Sibyll2.3d, QGSJETII-04, EPOS-LHC, and Pythia-Simple, respectively.

Figure 1a compares parton showers predictions to the CMS data at 0.9 TeV. It is clear that the Simple showers very well describe the data over the entire $p_{T}$ range, while Dire shower predictions are higher than experimental data, which could be due to the kinematics involved in Dire showers and may be due to the nPQCD effects. The Vincia shower predictions are 10% lower than the data at low $p_{T}$ values, but apparent underestimation is seen at high $p_{T}$ values, which goes up to 50%, possibly due to the angular ordering of partons. Figure 1c compares the predictions of the parton showers with the CMS data at 2.36 TeV. It is observed that the behavior of the distributions is the same as in the case of 0.9 TeV energy, while for the right side of the panel, the models’ predictions come close to experimental data. At $0<p_{T}<1$ GeV/c, all the models describe the CMS very well, while for the late $p_{T}$ region, deviation from data is observed, although it is less compared to 0.9 TeV energy. Figure 1e shows that the Dire shower explains the experimental data well over the entire $p_{T}$ range. In contrast, the Simple shower agrees well for up to $p_{T}=2.6$ GeV/c. In the later $p_{T}$ region, the models’ predictions deviate slightly from experimental data, which could be due to the multi-parton interactions.

While comparing the different modules of the Pythia8 model, we found that although the Dire model predicts better than the other two at 7 TeV, overall, the Simple module reproduced good results. Therefore, in the right panel of Figure 1, we used the predictions of the Simple module of the Pythia8 with Sibyll2.3d, QGSJETII-04, and EPOS-LHC for comparison to the experimental data. Figure 1b compares different models’ predictions with the CMS data at 0.9 TeV. One can see that all the models give good descriptions for $0<p_{T}<1$ GeV/c, while in later $p_{T}$ regions, all models underestimate the experimental data. The deviation is more in the case of the Sibyll2.3d event generator, while the EPOS-LHC and Pythia yielded a good comparison, which is within 10% in most of the cases throughout the $p_{T}$ range. Figure 1d compares the CRMC models and PYTHIA8 (Simple module) predictions with the experimental data. It is clear from the figure that the PYTHIA8 model describes the experimental data with 10% over the entire $p_{T}$ region, while all other models deviate from the experimental data by 30%. Figure 1f shows that the QGSJETII-04 model reproduces the data within 15 % over the entire $p_{T}$ range up to $p_{T}=1$ GeV/c. The Sibyll2.3d model has better prediction than the Pythia8 and EPOS-LHC model. The former has a deviation up to 30% while the latter has one up to 40%, mostly in the region of $p_{T}$; $1.5<p_{T}<5$, which may be due to parton fragmentation.

Figure 2 is the pseudorapidity distribution of primary charged hadrons within the −2.4<η<2.4 range integrated over transverse momentum at LHC energies. In Figure 2a,b (upper panels), at 0.9 TeV, the Simple showers agree very well with the experimental data, while Dire overestimates and Vincia underestimates the data. The observed deviations may be due to the angular ordering of partons. The PYTHIA8 predictions can also describe the experimental data over the entire η region. On the other hand, Sibyll2.3d, EPOS-LHC, and QGSJETII-04 underestimate the CMS data. Figure 2c,d (middle panels), at 2.36 TeV, shows that non of the parton showers, PYTHIA8, and CRMC models could explain the experimental data. Figure 2e,f (lower panel), at 7 TeV, shows that Dire and Vincia showers do not agree well with the experimental data. On the other hand, Simple shower data are slightly above the experimental data. It is clear from the figure that EPOS-LHC, QGSJETII-04, and Sibyll2.3d are lower than the experimental data, and the PYTHIA8 predictions are slightly higher than the experimental data.

### 3.2. Fit Procedure by Standard Distribution

We used the two-components standard distribution (Equation (Equation 4)) to fit the distributions of charged particles’ $p_{T}$ of the experimental data in the range of $p_{T}$, $0<p_{T}<4$ GeV/c and for pseudorapidity range of |η|<2.4 at 0.9, 2.36, and 7 TeV. The solid curves resulted from the fit procedure given in Figure 3. The markers with different colors show the experimental data, while lines with the same color as the corresponding data points show the fit results. We used a blue square, red circle, and orange triangle to show the data and a line with blue, red, and orange colors to show the fit curve at 0.9, 2.36, and 7 TeV. The parameters extracted from the fit function are given in Table 1. The $\chi^{2}/ndf$ values, given in the second last column of the table, show that the function fits the data well. The $T_{1}$ values are 0.163±0.008, 0.169±0.008, and 0.182±0.009 at 0.9, 2.36, and 7 TeV, respectively, while the respective values for $T_{2}$ are 0.389±0.012, 0.396±0.013, and 0.494±0.013. The value of *T*, calculated using Equation (Equation 5), varies as 0.20368±0.01, 0.22348±0.011, and 0.24128±0.012 for 0.9, 2.36, and 7 TeV, respectively, which shows a growth of *T* with increasing the center of mass energy. This is due to the fact that more energy is stored in the system as a result of a very violent reaction at higher energies, and correspondingly the system is excited to a high degree. This observation is harmonious with some of our previous results, such as [39,40,41,42], but it disagrees with other results, such as [43,44]. Finally, the parameter *N* has the values 360±18, 460±23, and 620±31 from the fit procedure for the data at 0.9, 2.36, and 7 TeV, hence showing an increasing trend with the rising energy. $N_{0}$ is a free parameter with a special significance in that it reveals the multiplicity. At higher energies, there is a harsh squeeze in the colliding system, where copious particles are produced. Therefore, larger $N_{0}$ at higher energies evinces the production of copious particles. It is pertinent to mention that $N_{0}$ is different from *C*. The constant *C* is used to normalize the integral of Equation (Equation 1), while the constant $N_{0}$ is used for the comparison of the fit function f(pT) and experimental data.

## 4. Summary and Conclusions

The pseudorapidity and transverse momentum distributions of charged hadrons are presented for the kinematic range of $0<p_{T}<4$ GeV/c and |η|<2.4 at 0.9, 2.36, and 7 TeV in pp collisions. The data of the CMS measurements are compared to simulation results using Monte Carlo event generators, i.e., PYTHIA8 and CRMC models. The there modules for parton showers in PYTHIA8 (Simple Showers, Vincia showers, and Dire showers) are analyzed. First, the predictions of the three modules are compared to experimental data, and we found that the Simple showers model yields the best results. In the second step, The prediction of the Simple showers module along with Sibyll2.3d, QGSJETII-04, and EPOS-LHC are compared to experimental measurements. Such studies are very useful to soft parton interactions and hadronization processes and tune event generators, including but not limited to, parton interaction, correlation, hadronization, spin, final state effects, etc. We record our findings of the comparison in the following:The Simple parton showers explain the CMS data very well for $p_{T}>1$ GeV/c at 0.9 and 2.36 TeV within the experimental errors, while Dire overshoots and Vicia undershoots the data by 50% each;At 7 TeV, the Dire module has a good prediction, whereas the Simple and Vincia modules underestimate the data within 30% and 50%;The comparison of the Simple module of the Pythia model, along with the predictions of the CRMC models with the experimental data, show that at 0.9 TeV, EPOS-LHC has better results than the others;At 2.36 GeV, CRMC models have better prediction than the Simple module of Pythia at low $p_{T}$, while QGSJETII-04 predicts well at high $p_{T}$. Both QGSJETII-04 and EPOS-LHC have closer results than the Pythia-Simple and Sibyll2.3d at 7 TeV;In the case of the pseudorapidity distributions, only the Pythia-Simple reproduced the experimental measurements at all energies. The Dire module overestimates while Vincia underestimates in decreasing order of discrepancy (20%, 12%, 5%) with energy;All CRMC models underestimate over the entire η range and for all three energies by 20%;Furthermore, A fit procedure using the standard distribution resulting in the $T_{eff}$ for experimental data shows that the $T_{eff}$ of the hadronic matter increases with an increase in the center of mass energy;The observed inconsistencies may be linked to the kinematics involved in the simulation of charged hadrons;In the case of parton showers, the angular ordering of partons could be the possible reason for the deviation;In contrast, the multi-parton interaction and parton fragmentation could be a possible reason in the case of CRMC models.

## Figures and Tables

**Figure 1 entropy-25-00452-f001:**
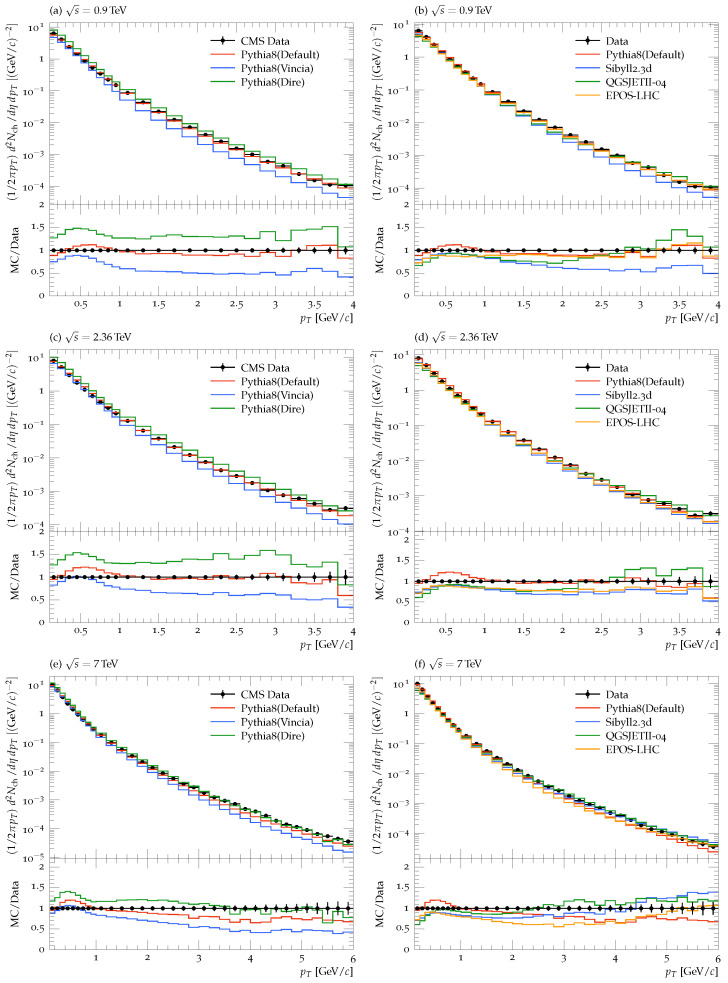
$p_{T}$ distribution of charged hadrons for |η|<2.4 by using Monte Carlo event generators. The Left panel shows the comparison of the different modules of Pythia8 with the experimental data at 0.9 (**a**), 2.36 (**c**), and 7 TeV (**e**) [4,5]. The right panel shows the predictions of the Pythia8 (Simple module only) and several CRMC models compared to the experimental data at the three different energies, 0.9 (**b**), 2.36 (**d**), and 7 TeV (**f**). The lower panel of each plot shows the ratio of the corresponding MC predictions to the experimental data (MC/Data).

**Figure 2 entropy-25-00452-f002:**
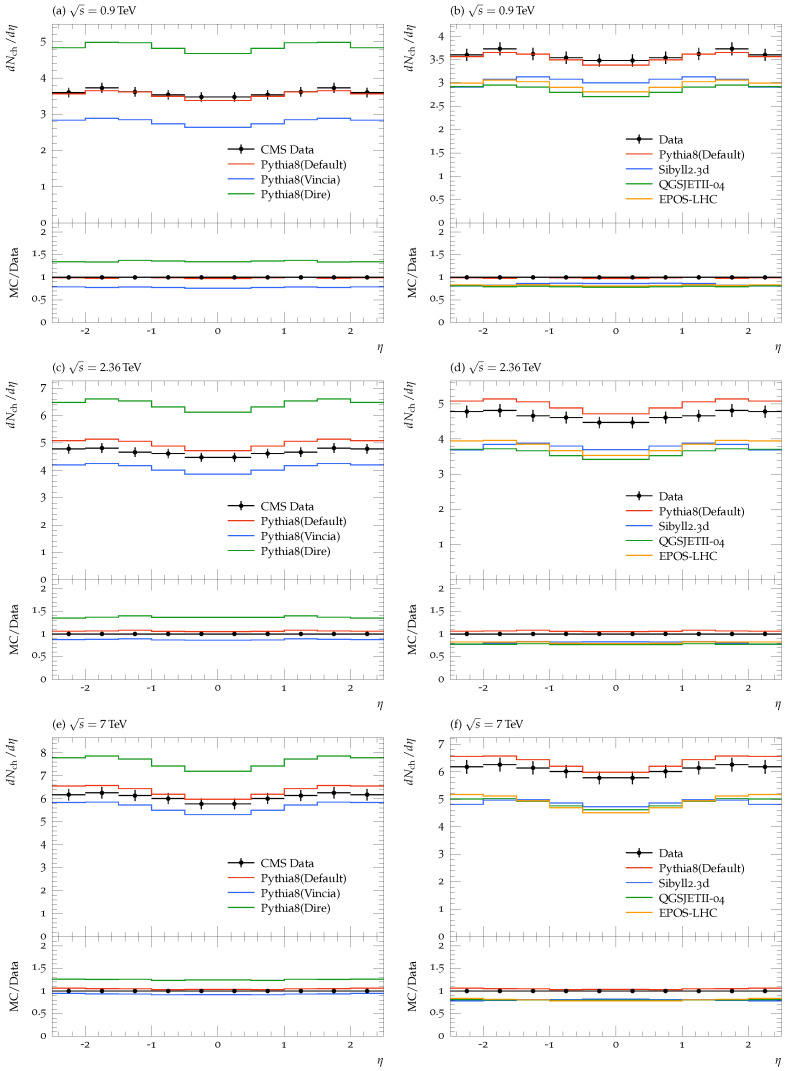
Pseudorapidity density of charged hadrons for η integrated over $p_{T}$ using PYTHIA8 parton showers and CRMC models at 0.9, 2.36, and 7 TeV [4,5]. The Left panel shows the comparison of the different modules of Pythia8 with the experimental data at 0.9 (**a**), 2.36 (**c**), and 7 TeV (**e**). The right panel shows the predictions of the Pythia8 (Simple module only) and several CRMC models compared to the experimental data at the three different energies, 0.9 (**b**), 2.36 (**d**), and 7 TeV (**f**). The ratios of the MC prediction to data measurements (MC/DATA) are plotted in the lower panel of each plot.

**Figure 3 entropy-25-00452-f003:**
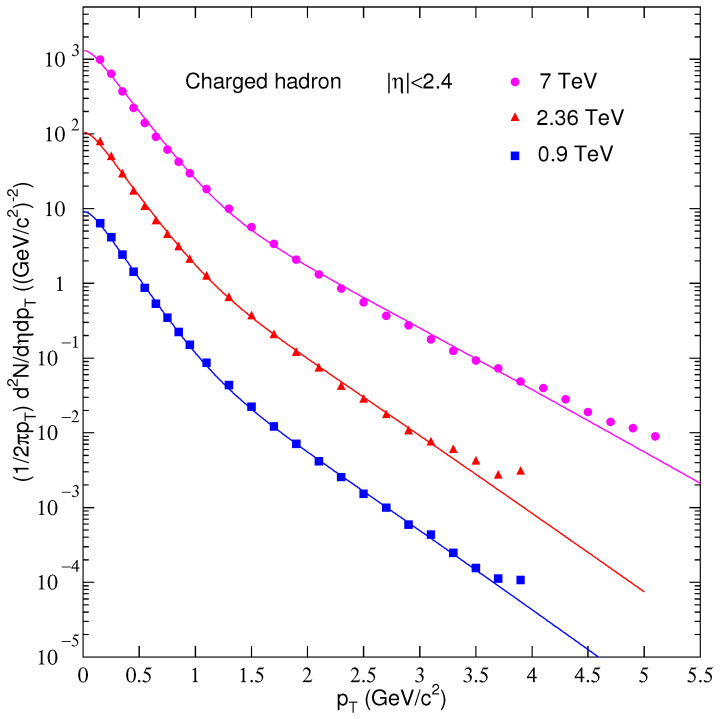
Pseudorapidity density of charged hadrons for η integrated over $p_{T}$ using PYTHIA8 parton showers and CRMC models at 0.9, 2.36, and 7 TeV, respectively. The MC simulation data are compared with CMS data. The MC to data ratio is plotted in each figure’s lower panel.

**Table 1 entropy-25-00452-t001:** Result obtained from the fit function using Equation (Equation 1) for the free parameters $T_{1}$, $T_{2}$, *T*, (*N*), ($\chi^{2}$), corresponding to the curves in Figure 1. The fit curves resulting from the fit function are given in Figure 3.

Energy	$T_{1}$ (GeV)	$T_{1_{\mathrm{error}}}$ (GeV)	$T_{2}$ (GeV)	$T_{2_{\mathrm{error}}}$ (GeV)	*z*	$z_{\mathrm{error}}$ (GeV)	*N*	$N_{\mathrm{error}}$	$\chi^{2}$/dof
0.9 TeV	0.163	0.008	0.389	0.012	0.82	0.01	360	18	6.6382
2.36 TeV	0.169	0.008	0.396	0.013	0.76	0.01	460	23	4.1403
7 TeV	0.182	0.009	0.494	0.013	0.81	0.01	620	31	6.0463

## Data Availability

Not applicable.

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
