# Peer review of "Charged Particles Transverse Momentum and Pseudorapidity Distribution in Hadronic Collisions at LHC Energies"

_entropy, 2023, doi:10.3390/e25030452_

Round 1

Reviewer 1 Report

Title: Charged particles transverse momentum and pseudorapidity distribution in hadronic collisions at LHC energies
===============================================================================

Comments to Author(s):

In this article, the author studies the pseudorapidity, η, and transverse momentum, pT, distributions of charged hadrons in pp collisions for the kinematic range of 0 < pT < 4 GeV/c and |η| < 2.4 at 0.9, 2.36 2 and 7 TeV.

The first part is concentrated on the comparison of different MC models, Pythia8 with three different parton shower models (Simple, Vincia, and Dire showers), Sibyll2.3d, QGSJETII-04, and EPOS-LHC to the experimental data obtained by CMS collaboration. In the second part, the effective temperature of the produced system is obtained by fitting the transverse momentum spectra by a two-component model with the assumption that both of them are in equilibrium.

The manuscript is well written and the results are also interesting, however, I have a few comments which need to be addressed before the publication.

Major comments:

1-
The author didn't mention how many MC events are generated for comparison to the data. It is better to show the uncertainties arising due to the MC models.

2-
What are the error bars on the solid black markers in the bottom panel of figure 1? Please consider both stat and systematic uncertainties for that.

3-
The Dire showers always overestimate the data. Do you have used used proper weight to normalise the spectra as mentioned in the Pythia manual.

https://pythia.org/latest-manual/PartonShowers.html

"The Dire shower comes with a nontrivial variable weight. It is therefore important that results for each event are weighted by the event weight in pythia.info.weight(), e.g. when filled in a histogram."

4-
What is the value of "z" while extracting the fit results? Please add this parameter to table 1.

Minor comments:

Line 4:  
EPOS-LHC-LHC -->  EPOS-LHC

Line 49:
mini bias --> minimum bias

Line 91:  
(y_{maximum}) -->  y_{maximum}

Line 50:

"Increasing the energy results in 150 increasing the yield; hence, the distributions get closer to the experimental"
The above sentence is not clear in the manuscript.

Line 157:
2.4 < η < 2.4 --> -2.4 < η < 2.4

Author Response

Dear respected reviewer,

Please find our responses in attachment.

Regards,

Authors,

Reviewer 2 Report

The author reports transverse momentum and pseudo-rapid distributions for pp collisions at LHC energies using several Monte Carlo event generators, including Pythia8, and compares them with experimental data obtained at CMS.

The predictions of each event generator were compared with the experimental data, and the results were reported in detail.

However, unfortunately, no comments or conclusions are clearly stated on what physics was revealed in pp collisions at LHC energies from the comparisons made by these event generators.

The authors should address the physics revealed by this study in  Sec.4, Summary and Conclusions.

Author Response

Dear respected reviewer,

Thank you very much for your review. Please find our responses in attachment. 

Regards,
Authors,

Reviewer 3 Report

The authors have a good comparison between the event generator simulation results and the CMS experimental data. The comparison between Pythia and cosmic rays MC (and also among the different modules of Pythia itself) is interesting and valuable. I would suggest the paper be published in its current form, with two minor changes: 1) in Page 2 line 54, change p_x^2 and p_y^2 to p_x and p_y since they are the momentum not the momentum square, 2) change all eq. to Eq. or change all Eq. to eq.